# Timing of Initiation of Renal Replacement Therapy in Sepsis-Associated Acute Kidney Injury

**DOI:** 10.3390/jcm9051413

**Published:** 2020-05-10

**Authors:** José Agapito Fonseca, Joana Gameiro, Filipe Marques, José António Lopes

**Affiliations:** Division of Nephrology and Renal Transplantation, Department of Medicine Centro Hospitalar Lisboa Norte, EPE Av. Prof. Egas Moniz, 1649-035 Lisboa, Portugal; joana.estrelagameiro@gmail.com (J.G.); filipedcmarques@campus.ul.pt (F.M.); jalopes93@hotmail.com (J.A.L.)

**Keywords:** acute kidney injury, prevention, renal replacement therapy, sepsis, treatment

## Abstract

Sepsis-associated acute kidney injury (SA-AKI) is a major issue in medical, surgical and intensive care settings and is an independent risk factor for increased mortality, as well as hospital length of stay and cost. SA-AKI encompasses a proper pathophysiology where renal and systemic inflammation play an essential role, surpassing the classic concept of acute tubular necrosis. No specific treatment has been defined yet, and renal replacement therapy (RRT) remains the cornerstone supportive therapy for the most severe cases. The timing to start RRT, however, remains controversial, with early and late strategies providing conflicting results. This article provides a comprehensive review on the available evidence on the timing to start RRT in patients with SA-AKI.

## 1. Introduction

Acute kidney injury (AKI) is a clinical syndrome characterized by a rapid decline in kidney function, encompassing numerous etiologies, and different and complex pathophysiological processes [1,2]. AKI is particularly important since it is strongly associated with increased hospital length of stay (LOS), costs, in-hospital mortality, development of chronic kidney disease (CKD) and long-term mortality risk [3,4,5,6]. The global burden of AKI accounts for around 13.3 million cases a year, and in the United States alone hospitalizations for AKI are on the rise [7,8,9].

The definition of AKI has undergone serious modifications over the last twenty years. Since the first consensual definition provided in 2004 by the Risk, Injury, Failure, Loss of kidney function and End-stage renal disease (RIFLE) classification [10], the initial term of acute renal failure (ARF) evolved to AKI as small increases in serum creatinine (SCr) were associated with increased mortality [1,11], widening the spectrum of the definition which now comprises from minor changes in markers of renal function to requirement for renal replacement therapy (RRT), as adopted by the Acute Kidney Injury Network (AKIN), in 2007 [12]. More recently, in 2012, the Kidney Disease: Improving Global Outcomes (KDIGO) combined both RIFLE and AKIN classifications, allowing for a definition that could be applied in clinical medicine, research and for public health purposes (Table 1) [13]. These definitions are not immutable and new definitions, chiefly concerning the duration of AKI, have recently been addressed, as observed by the proposed definitions for transient and persistent acute kidney disease (AKD) [14].

Acute complications related to AKI are diverse and depend on the severity of the insult [15]. Prompt recognition and early intervention directed to the cause of AKI and to the related complications, may soften both the duration and severity of this clinical syndrome, improving outcomes [16]. Special attention must be paid to identifying and eliminating potential insults, namely drugs, hypotensive states and exposure to iodinated contrast agents. Adequate volume assessment, as well as acid-based and electrolyte disturbances management must be addressed as well [17,18,19,20]. Nutritional support, surveillance of uremic symptoms and uremic bleeding associated with platelet dysfunction are also of critical importance in the management of AKI and the respective outcomes [21,22]. However, to date, no specific pharmacological therapy has proved to be effective in the prevention or treatment of AKI, partly owing to the various insults and different pathophysiological pathways associated with AKI [23].

In recent years, retrospective and prospective studies, as well as randomized controlled trials (RCTs), have focused on the timing of initiation of renal replacement therapy (RRT), yielding inconsistent results [24,25,26,27]. Most studies are considerably heterogenic concerning the definitions of early and late strategies. The clinical setting of the population is also highly variable between studies [28,29]. Whether an early, compared to a delayed strategy, for RRT initiation can improve outcomes in AKI is still a matter of debate.

Sepsis is the leading cause of AKI in critically ill patients, and sepsis-associated AKI (SA-AKI) is associated with higher severity scores, an increased need for RRT, and increases the risk of death and prolonged LOS compared to nonseptic AKI [30,31]. SA-AKI has a proper pathophysiology that exceeds the previously admitted acute tubular necrosis resulting from kidney ischemic injury. It is now understood that a set of different mechanisms including intra-renal and systemic inflammation, oxidative stress, microvascular and endothelial dysfunction are detrimental in the pathophysiology of SA-AKI [32,33,34].

Data regarding the impact of the timing of RRT initiation in SA-AKI are scarce and somewhat contradictory and inconclusive. In fact, most of the studies on this topic are small, retrospective, single-center and observational, and in studies addressing the impact of the timing of RRT initiation in the general ICU patients, data on SA-AKI patients are almost inexistent.

The aim of this review is to focus on the different studies comparing early versus delayed strategies in initiation of RRT in SA-AKI.

## 2. Timing of Initiation of RRT

Established and widely accepted indications for starting RRT include refractory fluid overload, severe hyperkalemia and metabolic acidosis refractory to medical therapy, signs of uremia (namely pericarditis and encephalopathy) and intoxications from dialyzable drugs or poisons [35].

There is no established universal definition for early initiation of RRT. Studies reporting different populations allocate to the “early” cohorts of patients with different serum urea levels, according to the RIFLE, AKIN or KDIGO classifications, with different urinary outputs and starting RRT within different times after AKI detection [28,36]. The potential benefits of earlier RRT initiation include premature acid-base and electrolyte equilibrium, avoidance of hypervolemia, which can be deleterious and provokes higher mortality according to different studies, elimination of uremic toxins and reducing both renal and systemic inflammation [37,38]. Retrospective and prospective cohorts, as well as RCTs, have proven the benefits of this approach, translated into earlier recovery of renal function, lesser hospital LOS and improved survival [36,39,40,41].

“Late” or “delayed“ initiation RRT is also loosely defined. Most studies also report more elevated serum urea or creatinine, a higher grade of AKI classified by the RIFLE, AKIN or KDIGO classifications, lower urinary output or even only formal indications for RRT initiation as criteria for the “late” group [28,36]. A delayed start of RRT may allow for patient stabilization, ensuring mainly hemodynamic and ventilatory conditions, reducing potential complications associated with prompt RRT start (catheter misplacement, catheter-related bloodstream infections, bleeding or thrombotic events), and even permitting time to renal recovery averting the need for RRT [42,43,44]. Despite great enthusiasm for an early start of RRT, more recent RCTs and meta-analysis have questioned this approach, reporting no survival advantage [24,25,28,45].

Whether these principles also apply exclusively to SA-AKI is also unknown. Table 2 summarizes the different studies that addressed early vs. late initiation of RRT in SA-AKI cohorts.

A Korean retrospective single-center study with 60 patients assessed whether a shorter interval between the start of early goal-directed therapy (EGDT) and CRRT initiation was an independent predictor of mortality in patients with SA-AKI. Patients were divided into an early (≤26.4 h) CRRT initiation group and a late (>26.4 h) CRRT initiation group. All-cause mortality at 28 days was significantly higher in the late CRRT group (30.0% in the early group vs. 56.7% in the late group), even after adjusting for diabetes mellitus, liver failure, and Acute Physiology and Chronic Health Evaluation II scores, conferring a potential benefit for early initiation of CRRT [46].

Earlier, an observational retrospective single-center study with 55 SA-AKI patients at the injury or failure stages of the RIFLE classification compared outcomes between inception of early (≤24 h) and late (>24 h) continuous renal replacement therapy (CRRT). The primary outcome, 28-day mortality, was lower in the early group (19.4% vs. 47.4%) but no differences were observed between the injury or failure stages [47].

A retrospective analysis of time from AKI onset to CRRT initiation was performed according to ICU mortality in 158 septic shock patients with AKI. Mortality rate was high at ICU discharge (50.6%), and non-survivors initiated CRRT later than survivors with a cut-off time from AKI onset to CRRT initiation for ICU mortality of 16.5 h. The cumulative mortality rate was significantly higher in patients in whom CRRT was initiated beyond 16.5 h after AKI onset than in those in whom CCRT was initiated within 16.5 h, pointing to earlier initiation as a predictor of survival [48].

The definitions of “early” and “late” may also encompass other features than solely time from detection of AKI. A retrospective cohort study investigated 3-time interval parameters on the morbidity and mortality of 177 patients with septic shock–induced acute kidney injury. Time from vasopressor initiation to CRRT initiation < 24 h, but not time from ICU admission to CRRT initiation, nor time from endotracheal intubation to CRRT initiation was associated with survival and acted as an independent factor associated with 28-day and 90-day mortality. Therefore, time from vasopressor initiation to CRRT initiation < 24 h could be considered as a bundle for the definition of early CRRT initiation [49].

A retrospective cohort study of 130 ICU patients with sepsis and acute renal failure requiring RRT assessed whether early implementation of RRT (defined as a BUN < 100 mg/dL) compared to a late RRT initiation (after BUN ≥ 100 mg/dL) had an effect on the 28-day mortality. Survival rates were 67%, 47.7% and 30.7% at 14, 28 and 365 days, respectively, in the early group. Survival rates were 46.7%, 31.7% and 13.3%, respectively, in the late group. After a logistic regression analysis, initiating dialysis with a BUN > 100 mg/dL predicted mortality at 14 days and 365 days, thus early initiation of dialysis was associated with improved survival rates up to one year [50].

A retrospective analysis of 160 critically-ill patients with SA-AKI treated with or without CRRT was performed to investigate which AKI stage might be the optimal timing for starting CRRT. Starting CRRT at AKI stage 2 was associated with reduction in the 28-day mortality, increase in the 28-day survival rate, lesser ICU LOS and a reduction in ventilation time, compared with those in the control group. This hypothesizes that AKI stage 2 may be the ideal time to initiate RRT [51].

The first RCT goes back to 2009, when twelve French ICUs conducted a RCT with 80 patients enrolled within 24 h of developing the first organ failure related to a new septic insult. Patients were assigned to an early intervention group, who received hemofiltration (25 mL/kg/h) for a 96-h period, or to the conventional group, who received standard sepsis management. The primary end point (number, severity, and duration of organ failures during the 14 days) was significantly higher in the early intervention group, suggesting that early application of CRRT may be deleterious in severe sepsis and septic shock [52].

A retrospective single-center cohort study with 120 patients with septic shock and AKI evaluated the impact of early versus late initiation of CRRT on organ dysfunction. Patients were dichotomized into “early” (stage Risk of the RIFLE classification) or “late” (stages Injury or Failure of the RIFLE classification) CRRT initiation. Organ dysfunction at 48 h, evaluated through the SOFA/non-renal SOFA, dialysis requirement and mortality at 28 days, 3 months and 6 months were similar between groups and no clinical benefits of early CRRT initiation were identified [53].

Chou and colleagues evaluated 370 septic patients presenting with AKI requiring RRT in a surgical ICU setting. Patients were divided into early (RIFLE-0 or -Risk) or late (RIFLE-Injury or -Failure) initiation of RRT by RIFLE criteria. Mortality rates in early (70.8%) and late (69.7%) RRT groups were similar and an early strategy was not associated to a decrease in hospital mortality. Therefore, the RIFLE classification was not able to predict the benefits of early RRT initiation in the context of septic AKI [54].

Finally, the Initiation of Dialysis Early versus Late in the Intensive Care Unit (IDEAL-ICU) was a multicenter RCT of 488 patients with septic shock and severe AKI at the failure stage of the RIFLE classification. The early group was defined as RRT start within 12 h of achieving the Failure stage without life-threatening AKI complications, and the delayed group started RRT after a delay of 48 h of achieving the Failure stage, if renal function recovery did not ensue. This trial demonstrated no significant difference in mortality between groups at 90 days [55].

## 3. Limitations

Several limitations have been pointed to the potential benefit of an early versus late strategy for RRT initiation. Results either showing advantage of an early strategy [27] or benefit from a more conservative approach [24,25,26] must be carefully interpreted since the ICU setting of these cohorts may be predominantly surgical [27,56]. Most studies include different population cohorts with clinical heterogeneities and mixed causes of AKI, in which the extrapolation of the results for SA-AKI is discouraged [24,25,26,27,57]. In addition, previous studies mostly combined early start with more-intensive dialysis and late start with less-intensive dialysis, creating a bias [13].

A recent and comprehensive meta-analysis concerning time of initiation of RRT in AKI described 18 RCTs, of which only three addressed exclusively SA-AKI cohorts and only one of these included more than 100 patients [58]. Study design, including small sample sizes, single-center and retrospective studies, paucity of double-blinded studies, heterogeneity in eligibility criteria, lack of high-quality studies as well as scarcity of RCTs pose major challenges in ascertaining the benefits from early versus delayed strategies [46,47,48,49,51,53].

RRT modalities (intermittent hemodialysis or CRRT) and dialysis dose have shown great disparity between studies, and were mostly left to clinicians’ decision, which creates a bias concerning patients’ outcomes [46,48,49,54].

AKI was also subject of different definitions, and not every study included urinary output as criteria to define AKI, thus leading to exclusion of potential patients with SA-AKI [53,54]. Most studies also report the RIFLE classification for AKI, whose sensitivity for detection AKI is known to be inferior compared to contemporary KDIGO classification [53,55,59,60].

Furthermore, the definitions of “early” and “late” are not standardized and remain to be defined according to clinicians’ criteria. These criteria may include time, hemodynamic parameters, biochemical characteristics, and what one study may define as “early” can be defined as “late” in other studies [28,58,61]. Additionally, criticism in the choice of delaying only 48 h to allow for renal recovery and including these patients in the “late” or “delayed” strategy has been stated. Kidney function may ensue much later than 48 h, allowing for latter renal recovery and avert the need for RRT [55].

Follow-up length is also inconsistent and despite some studies reporting evidence of the 28-day survival benefit for the early RRT group, the effect of this strategy on long-term survival remains unclear [47,49,61]. Moreover, the benefit of RRT might be attributable not only to the early nonspecific removal of inflammatory mediators but also to an early stabilization of the hemodynamic, respiratory, and biological status [46].

## 4. Future Perspectives

The optimal timing for initiation of RRT is not clarified on the basis of research evaluated to date both in SA-AKI and other etiologies of AKI [28,58,61]. The results of the studies included in our analysis are contradictory. Large, multicenter prospective observational studies are required to make sure the impact of CRRT timing on septic AKI [62].

To our knowledge, no specific RCT is underway to ascertain the optimal time for RRT initiation in SA-AKI. The Standard Versus Accelerated Initiation of Dialysis in Acute Kidney Injury (STARRT-AKI) is an ongoing multicenter trial that plans to enroll at least 2866 critically ill patients in order to detect a 6% difference in mortality in favor of an early RRT start strategy. A specific subgroup analysis for patients with sepsis and septic shock, as defined by the Sepsis-3 criteria based on the rationale that earlier RRT, due to more aggressive removal of inflammatory mediators, might have a more prominent effect among patients with SA-AKI, is awaited in order to define whether an early strategy could be beneficial for this specific population [63].

All in all, timing to start RRT should be individualized to patients’ clinical status and laboratory progression. Nonetheless, an early RRT start in septic patients could potentially improve outcomes by limiting systemic inflammation, fluid overload and organ injury, though consistent evidence is still lacking. Moreover, SA-AKI is associated with lower SCr due to reduced production of creatinine and hemodilution associated with a considerably positive fluid balance, and a more pronounced oliguria, thus a less severe KDIGO stage defined these criteria might underestimate the severity of AKI and create a bias in defining RRT timing. Novel biomarkers, such as NGAL and TIMP-2 x IGFBP-7, have been reported as potential predictors of AKI severity after admission, which might prove useful in deciding the timing to start RRT in this setting, though further studies are still required to validate the routine use of these biomarkers in clinical practice.

## 5. Conclusions

In conclusion, the available evidence demonstrates contradictory results concerning the benefits of early RRT initiation in SA-AKI. Moreover, postponing RRT initiation may allow for renal recovery from AKI, averting the need for RRT. Evidence results essentially from small, retrospective, non-double blinded studies and although evidence in mortality improvement is lacking, especially with RCTs, other outcomes such as ICU and hospital LOS, development of CKD and late mortality were rarely assessed. Definitions for “early” and “late” start need to be standardized and the optimal time to initiate RRT remains undefined. Large multicenter randomized trials with better design are needed to answer these questions.

## Figures and Tables

**Table 1 jcm-09-01413-t001:** Acute kidney injury according to the Kidney Disease: Improving Global Outcomes (KDIGO) classification. SCr—serum creatinine, UO—urinary output.

Acute Kidney Injury Staging According to Kidney Disease Improving Global Outcomes (KDIGO) Classification		
**Stage**	**SCr**	**UO**
1	↑ SCr ≥ 26.5 μmol/l (≥ 0.3 mg/dl) or ↑ SCr ≥ 150–200% (1.5–1.9×)	<0.5 mL/kg/h (>12 h)
2	↑ SCr > 200–300% (> 2–2.9×)	<0.5 mL/kg/h (>12 h)
3	↑ SCr > 300% (≥3×) or ↑ SCr to ≥ 353.6 μmol/l (≥4 mg/dl) or initiation of renal replacement therapy	<0.3 mL/kg/h (24 h) or anuria (12 h)

**Table 2 jcm-09-01413-t002:** Trial summary characteristics reporting data on early vs. late renal replacement therapy initiation.

Study	Design	N	RRT Modality	Early RRT Start	Late RRT Start	Follow-Up	Mortality Early vs. Late RRT Start	AUROC
Baek, 2017	Retrospective, single-center, cohort study	177	CRRT	initiation within 24 h of vasopressor treatment (Tvaso-CRRT less than 24 h)	initiation beyond 24 h of vasopressor treatment (Tvaso-CRRT over 24 h)	28 days, 90 days	28 days - 33.6% vs. 61.5% (p = 0.001) adjusted OR 0.449 (95% CI 0.211–0.956), p = 0.038 90 days - 44.0% vs. 75.0% (p < 0.001) adjusted OR 0.369 (95% CI 0.165–0.825), p = 0.015	Tvaso-CRRT >24 h, AUC, 0.634; 95% CI, 0.559–0.705, p = 0.001;
Barbar, 2018	Multicenter, RCT	488	RRT	RRT within 12 h after documentation of failure-stage AKI	RRT after 48 h if renal recovery had not occurred	28 days, 90 days, 180 days	28 days - 45% vs. 42% (p = 0.48) 90 days - 58% vs. 54% (p = 0.38) 180 days - 61% vs. 57% (p = 0.37)	-
Carl, 2010	Retrospective, single-center, cohort study	147	RRT	BUN < 100 mg/dL + AKIN stage ≥ 2	BUN ≥ 100 mg/dL + AKIN stage ≥ 2	14 days, 28 days, 365 days	14 days - 33% vs. 53.3% (p = 0.01) adjusted OR 3.6 (95% CI 1.7–7.6), p = 0.001 28 days - 52.3% vs. 68.3% (p < 0.05) adjusted OR 2.6 (95% CI 1.2–5.7), p = 0.01 365 days - 69.3% vs. 86.7% (p < 0.05) adjusted OR 3.5 (95% CI 1.2–10), p = 0.02	-
Chon, 2012	Retrospective, single-center, cohort study	55	CRRT	≤24 h (mean time to RRT = 12.5 h) RIFLE-I and RIFLE-F	>24 h (mean time to RRT = 42.2 h) RIFLE-I and RIFLE-F	28 days, 90 days	28 days - 19.4% vs. 47.4% (p = 0.030) adjusted HR 3.378 (95% CI 1.174–9.722), p = 0.024 90 days - 38.2% vs. 61.1% (p = 0.115)	-
Chou, 2011	Retrospective, single-center, cohort study	370	RRT	sRIFLE-0 or -Risk	sRIFLE-Injury or -Failure	during ICU stay	70.8% vs. 69.7%, p = 0.98	-
Oh, 2016	Retrospective, single-center cohort study	60	CRRT	≤26.4 h mean time between EGDT and CRRT initiation 7.9 h (1.0–25.1 h)	>26.4 h mean time between EGDT and CRRT initiation 61.5 h (32.3–137.6 h)	28 days	30.0% vs. 56.7%, p = 0.037 Late CRRT treatment (vs. early CRRT treatment) adjusted HR 2.461 (95% CI 1.044–5.800), p = 0.040	-
Payen, 2009	Prospective, randomized, multicenter study	76	CRRT	RRT for at least 96 h within 24 h of randomization,	No RRT unless metabolic renal failure and classic indications for RRT present	28 days	CRRT vs. control (54% vs. 44%; p < 0.49)	-
Shum, 2013	Retrospective, single center, cohort study	120	CRRT	simplified RIFLE-Risk (Mean time from ICU admission to RRT = 20.7 h)	simplified RIFLE-Injury or Failure (Mean time from ICU admission to RRT = 10.8 h)	28 days, 3 months and 6 months	28 days - 48.4% vs. 48.3% (p = 0.994) 3 months - 58.1% vs. 55.1% (p = 0.771) 6 months - 61.3% vs. 56.2% (p = 0.62)	-
Tian, 2014	Retrospective, single center, cohort study	160	CRRT	CRRT group	control group	28 days	AKIN 1 - 21.7% vs. 42.3% (NS) AKIN 2 - 38.7% vs. 66.7% (p = 0.048) AKIN 3 - 67.4% vs. 84.6% (NS) adjusted OR 0.254 (95% CI 0.072–0.897), p = 0.033	
Yoon, 2018	Retrospective, single center, cohort study	158	CRRT	<16.5 h	≥16.5 h	28 days, 60 days, 90 days	28 days - 40.7% vs. 70.8% HR 2.118 (95% CI 1.375–3.261), p < 0.001 60 days - HR 2.244 (95% CI 1.497–3.363), p < 0.001 90 days - HR 2.115 (95% CI 1.424–3.141), p < 0.001 Interval time from AKI to CRRT initiation Adjusted HR 1.016 (95% CI 1.008–1.025; p < 0.001)	interval time from AKI to CRRT initiation for ICU mortality AUC 0.786 (95% CI, 0.716–0.856; p < 0.001)

AKI—acute kidney injury, AKIN—Acute Kidney Injury Network, AUC—area under curve, AUROC—area under receiver operating characteristic, BUN—blood urea nitrogen, CI—confidence interval, CRRT—continuous renal replacement therapy, EGDT—early goal directed therapy, HR—hazard ratio, ICU—intensive care unit, RIFLE—Risk, Injury, Failure, Loss of kidney function and End-stage renal disease, OR—odds ratio, RCT—randomized controlled trial, RRT—renal replacement therapy, Tvaso—time from vasopressor.

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
