# Peer review of "Timing of Initiation of Renal Replacement Therapy in Sepsis-Associated Acute Kidney Injury"

_jcm, 2020, doi:10.3390/jcm9051413_

Round 1

Reviewer 1 Report

The authors discussed about the timing of RRT in septic KI. The issue is important and the reference they reviewed is sufficient, I feel.

However, throughout the reading of this paper, I could not get any suggestion for clinical decision making.

I feel, at least, the author should define their standpoint whether pros or cons for the discussed issue.

In addition, the authors discussed about the ambiguity of the definition of early and late intervention, the reviewer deeply understand the situation at now. But I hope the authors show any direction whether which specific timing is the best  at least the authors believe, and its evidence based explanation. 

Again, to accept this review finally, I want the authors to establish their standpoints whether the early or late is better, and which specific timing should be used as definition, rather than mere collecting the conflicting evidences and conclude that the condition is controversial.

Author Response

Dear Reviewer,

thank you for your comments and suggestions.

We have addressed your suggestions and added to the conclusions (lines 117-126, in light blue) our standpoint, specifying that we believe early RRT start in SA-AKI probably has a role concerning the damage cause by inflammation and some pitfalls related to serum creatinine concentration and oliguria.

Reviewer 2 Report

This topic has been well discussed in previous publications. Authors have discussed in detail the limitations of the current studies which is important from a clinical perspective on a widely debated topic. Below are few revisions recommended.

  1. Need to keep fond the same everywhere. Abstract itself has 2 fonds. 
  2. Please insert in First paragraph of introduction below sentence.

    "Global burden of AKI accounts for around 13.3 million cases a year and in United states alone hospitalizations for Acute kidney injury are on the rise over time". 

    References for the above sentence are. 

    a)Thongprayoon C, Kaewput W, Thamcharoen N, et al. Incidence and Impact of Acute Kidney Injury after Liver Transplantation: A Meta-Analysis. J Clin Med. 2019;8(3):372. Published 2019 Mar 17. doi:10.3390/jcm8030372

    b)Lertjitbanjong P, Thongprayoon C, Cheungpasitporn W, et al. Acute Kidney Injury after Lung Transplantation: A Systematic Review and Meta-Analysis. J Clin Med. 2019;8(10):1713. Published 2019 Oct 17. doi:10.3390/jcm8101713

           c)Thongprayoon C, Kaewput W, Thamcharoen N, et al. Acute Kidney Injury in    Patients Undergoing Total Hip Arthroplasty: A Systematic Review and Meta-Analysis. J Clin Med. 2019;8(1):66. Published 2019 Jan 9. doi:10.3390/jcm8010066

  3. Sentence 90-94 needs restructuring in the section of Timing of RRT.

  4. Sentence 98: Follow up length is also "inconstant". Did you mean inconsistent. 

  5. Sentence 119-121 in conclusion needs grammatical restructuring. 

  6. In conclusion section authors need to add that "Definitions for early and late start need to be standardized". 

Author Response

Dear reviewer,

thank you for your comments and suggestions.

I have reviewed the manuscript according to your indications.

Yours sincerely

Round 2

Reviewer 1 Report

Except the specification of late and early.....the author well responded to my inquiry.